# NSAIDs Induce Proline Dehydrogenase/Proline Oxidase-Dependent and Independent Apoptosis in MCF7 Breast Cancer Cells

**DOI:** 10.3390/ijms23073813

**Published:** 2022-03-30

**Authors:** Adam Kazberuk, Magda Chalecka, Jerzy Palka, Katarzyna Bielawska, Arkadiusz Surazynski

**Affiliations:** 1Department of Medicinal Chemistry, Medical University of Bialystok, Mickiewicza 2D, 15-222 Bialystok, Poland; kadam568@gmail.com (A.K.); magda.chalecka@umb.edu.pl (M.C.); pal@umb.edu.pl (J.P.); 2Department of Analysis and Bioanalysis of Medicines, Medical University of Bialystok, Mickiewicza 2D, 15-222 Bialystok, Poland; katarzyna.bielawska@umb.edu.pl

**Keywords:** mitochondria, proline metabolism, proline oxidase, proline dehydrogenase, NSAIDs, PPAR, COX, apoptosis, breast cancer, oxidative stress

## Abstract

Non-steroidal anti-inflammatory drugs (NSAIDs) are considered in cancer therapy for their inhibitory effect on cyclooxygenase-2 (COX-2), which is overexpressed in most cancers. However, we found that NSAIDs as ligands of peroxisome proliferator-activated receptor-γ (PPARγ)-induced apoptosis independent of the COX-2 inhibition, and the process was mediated through activation of proline dehydrogenase/proline oxidase (PRODH/POX)-dependent generation of reactive oxygen species (ROS). This mitochondrial enzyme converts proline to ∆1-pyrroline-5-carboxylate (P5C) during which ATP or ROS is generated. To confirm the role of PRODH/POX in the mechanism of NSAID-induced apoptosis we obtained an MCF7 CRISPR/Cas9 PRODH/POX knockout breast cancer cell model (MCF7^POK-KO^). Interestingly, the studied NSAIDs (indomethacin and diclofenac) in MCF7^POK-KO^ cells contributed to a more pronounced pro-apoptotic phenotype of the cells than in PRODH/POX-expressing MCF7 cells. The observed effect was independent of ROS generation, but it was related to the energetic disturbances in the cells as shown by an increase in the expression of AMPKα (sensor of cell energy status), GLUD1/2 (proline producing enzyme from glutamate), prolidase (proline releasing enzyme), PPARδ (growth supporting transcription factor) and a decrease in the expression of proline cycle enzymes (PYCR1, PYCRL), mammalian target of rapamycin (mTOR), and collagen biosynthesis (the main proline utilizing process). The data provide evidence that the studied NSAIDs induce PRODH/POX-dependent and independent apoptosis in MCF7 breast cancer cells.

## 1. Introduction

Cancer treatment strategies include different classes of drugs that are not typically antineoplastic. One class is the non-steroidal anti-inflammatory drugs (NSAIDs) known to decrease the risk of various cancers such as prostate, colorectal, breast, and lung [1,2,3]. However, NSAID-induced antineoplastic pathways are not well recognized. It has been considered that a potent target of NSAIDs in cancer therapy is inflammation [4]. Numerous studies have shown that an inflammatory environment facilitates cancer development. Cyclooxygenases (COXs) are key inflammatory regulators responsible for prostaglandin synthesis from fatty acids (such as linoleic and arachidonic acid). Two isoforms of this enzyme are known: constitutively expressed COX1 and conditionally expressed COX2 in response to inflammation. An important fact is that in most cancers COX2 is overexpressed [5], which is linked to regulation of invasiveness, angiogenesis, proliferation, and migration [6,7]. Although the hypothesis of the anticancer properties of NSAIDs was proven in vitro and in vivo (as they are COX2 inhibitors), other research has showed that in cancer cells lacking COX2 or in the cells with knocked-down expression of COX2, NSAIDs had similar anticancer activity. This suggests other COX2 independent anticancer activities of NSAIDs [8,9,10].

Indomethacin and diclofenac are representative NSAIDs and well-documented synthetic ligands of peroxisome proliferator-activated receptor-γ (PPARγ) [11]. It is known that three isoforms of PPAR exist: PPARα, PPARβ/δ, and PPARγ, and all of them have transcriptional activity as they belong to the nuclear receptor family. The main role of these receptors is to regulate adipogenesis, lipid metabolism, and glucose homeostasis, but they are also involved in the processes such as inflammation, proliferation, and cancer metabolism [12,13]. Ligands of these receptors are mostly synthetic drugs, as mentioned earlier, but also include natural ligands such as fatty acids or prostaglandins [14,15,16]. Thiazolidinediones (i.e., troglitazone, pioglitazone), the class of drugs used for type 2 diabetes mellitus treatment, are the most potent inducers of PPARγ. Cancer cells treated with thiazolidinediones evoke pro-apoptotic phenotypes, suggesting that the process is PPARγ-dependent [17,18,19].

Activation of PPARγ induces proline dehydrogenase/proline oxidase (PRODH/POX); (PRODH<, GenBank^TM^ NM_016335), the inner mitochondrial membrane flavin-dependent enzyme [20,21,22]. PRODH/POX catalyzes conversion of proline to ∆^1^-pyrroline-5-carboxylate (P5C). During this reaction free electrons are transported to the electron transport chain to produce pro-survival ATP or they are accepted by oxygen to generate reactive oxygen species (ROS) inducing apoptosis/autophagy [21,23,24,25]. However, the mechanism for driving PRODH/POX into a pro-survival or pro-apoptotic pathway is not clear. Conversion of proline into P5C is a part of proline turnover coupled with a reverse reaction in which P5C is turned into proline by three isoenzymes of P5C reductase (PYCR). Two of them are mitochondrial (PYCR1 and PYCR2) and PYCRL is localized in the cytoplasm. P5C reduction to proline and proline shuttling between mitochondria and cytosol are coupled to the glucose metabolism by the pentose phosphate pathway [26,27]. P5C is also a substrate for P5C dehydrogenase (P5CDH)-producing glutamate, which is a precursor of α-ketoglutaric acid, an important compound of the TCA cycle [21]. Thus, the proline–P5C cycle is an important metabolic regulator of the PRODH/POX-dependent pathway. Interestingly, COX-2, MAPK, EGFR, and Wnt/β-catenin signaling pathways are also related to PRODH/POX [28,29,30]. A potent inducer of PRODH/POX is tumor suppressor p53 [31]. Transcriptional regulation of PRODH/POX by p53 was found in the PRODH/POX promoter, containing a p53-response element [32].

PRODH/POX-induced functions are dependent on proline availability. The amino acid can come from protein degradation products, mainly derived from collagen. The final collagen degradation is catalyzed by prolidase.

Prolidase [E.C.3.4.13.9], known also as peptidase D or iminopeptidase, is a cytoplasmic imido-dipeptidase or imido-tripeptidase [33,34] that cleaves imido-peptides with C-terminal proline or hydroxyproline [35]. The substrate for prolidase is derived mainly from collagen degradation (the most abundant protein containing imino bonds) and also from other degraded proline-containing proteins [36,37]. Prolidase activity was found to recycle proline for collagen re-synthesis, and therefore, the enzyme may play an important role in the regulation of collagen biosynthesis. It has been well documented in fibroblasts treated with anti-inflammatory drugs [38], P5C [39], during experimental fibroblast aging [40], experimental chondrocyte inflammation [41], activation of integrin receptor for type I collagen [42], in fibroblast derived from Osteogenesis Imperfecta patients [43], and in several cancer tissues [44,45,46].

Other factors that contribute to cytoplasmic proline levels are proline utilizing processes. The most effective is collagen synthesis [47] that, apart from important biological functions, may work as a “sink” for proline. From the perspective of redox balance, collagen can also be a sink for reducing the potential of proline and removing it from the metabolic pool. In fact, metabolic abnormalities concomitant to inflammatory processes are frequently associated with collagen formation, i.e., fibrosis [48,49]. During the inhibition of collagen biosynthesis, reducing the potential of proline is carried out in mitochondria by PRODH/POX, yielding P5C. The intensity of this process is regulated by the energetic status of the cells, which is controlled by mTOR, AMPK (77–79), as well as certain oncogenes, e.g., c-Myc [50].

The aim of the study was to establish the role of PRODH/POX in NSAID-induced apoptosis in breast cancer cells. CRISPR/Cas9 technology [51,52,53,54] was used to prepare the MCF7 cell line with PRODH/POX knockout. In this article we provide evidence that indomethacin (IND) and diclofenac (DCF) induce apoptosis in both a PRODH/POX-dependent and a PRODH/POX-independent manner through different mechanisms.

## 2. Results

Breast cancer MCF7 and PRODH/POX CRISPR/Cas9 knockout MCF7 cells (MCF7^POX-KO^) were treated with 0.5 mM indomethacin and 0.375 mM diclofenac for 24 h to recognize the role of PRODH/POX in NSAID-dependent apoptosis. The viability of the cells was evaluated using the MTT method. As shown in Figure 1A, 24 h incubation of both cell lines with indomethacin (IND) and diclofenac (DCF) contributed to decreasing cell viability to 65 and 68% in MCF7 cells and to 24 and 27% in MCF7^POK-KO^ cells, respectively, compared to controls. This result showed that the studied NSAIDs evoked a stronger cytotoxicity in MCF7^POK-KO^ cells than in PRODH/POX-expressing MCF7 cells.

The data were corroborated by a drastic decrease in DNA biosynthesis in the NSAID-treated MCF7^POK-KO^ cells (Figure 1B). In IND- and DCF-treated MCF7 cells DNA biosynthesis was decreased to 51 and 48%, while in MCF7^POX-KO^ cells the process was decreased to 19 and 14% of control, respectively.

Interestingly, a strong inhibitory effect of the studied drugs on collagen biosynthesis was also observed, as shown in Figure 1C. In MCF7 cells treated with IND and DCF, collagen biosynthesis was decreased to 31 and 20% of control, respectively. In MCF7^POX-KO^ cells almost total inhibition of this process was found at 8 and 6% of control, respectively. Moreover, augmented inhibition of collagen biosynthesis (proline utilization process) in MCF7^POX-KO^ cells was also accompanied by a doubling of intracellular proline concentration (Figure 1D).

A cytometric assay with annexin V and propidium iodide was used to evaluate the proportion of healthy, early apoptotic, and late apoptotic (dead) cells in the NSAID-treated MCF7 and MCF7^POK-KO^ cells. In IND-treated MCF7 cells 36% of cells were healthy, 36% were early apoptotic, and 28% were late apoptotic (dead). MCF7 cells treated with DCF were in proportions of 54, 24, and 21%, respectively. In control (untreated) MCF7 cells, 88% of cells were healthy, 11% were early apoptotic, and only 1% were late apoptotic. Interestingly, in untreated MCF7^POK-KO^ cells, 61% of cells were healthy, 27% early apoptotic, and 12% were late apoptotic. This high number of early apoptotic and dead cells in the MCF7^POK-KO^ control might underline the pro-survival role of PRODH/POX in cancer cells. However, in MCF7^POK-KO^ cells treated with IND or DCF, cell proportions were: 13% healthy, 34% early apoptotic, 52% dead, and 16% healthy, 57% early apoptotic, and 27% dead, respectively (Figure 2).

It was previously shown that IND and DCF activate PPARγ, leading to PRODH/POX-dependent ROS-mediated apoptosis [20]. Both drugs were used to validate their ability to generate ROS. An experiment with 2′,7′-dichlorofluorescin diacetate staining confirmed that incubation of MCF7 cells with NSAIDs strongly increased ROS generation, as visualized by the increase in red fluorescence, compared to control (Figure 3). This effect was not shown in MCF7^POX-KO^ cells, suggesting the critical role of PRODH/POX in NSAID-dependent ROS generation.

However, Western blot analysis for apoptosis markers (Figure 4) in IND and DCF-treated MCF7 and MCF7^POX-KO^ cells showed an increase in expression of active caspase 7 (execution caspase) and caspase 9 (marker of the mitochondrial apoptotic pathway). Interestingly, expression of caspase 8 (apoptotic marker of the extrinsic pathway) was increased in NSAID-treated MCF7^POX-KO^ cells, while in MCF7 cells the expression was not affected (Figure 4). This suggested that the PRODH/POX knockout promoted the extrinsic apoptosis activation pathway. Elevated expression of pro-apoptotic BID in MCF7^POX-KO^ cells compared to MCF7 suggested downstream regulation of apoptosis via caspase 8 activation. The significant difference in expressions of total and cleaved PARP between the studied cells (weak expression in MCF7^POX-KO^ cells) suggested that in cells with functional PRODH/POX, NSAID-induced apoptosis is associated with DNA damage in response to environmental stress (such as oxidative stress). Evaluation of autophagy marker expression such as Beclin1, ATG7, and ATG5 showed that in MCF7 and MCF7^POX-KO^ cells treated with NSAIDs autophagy is not involved. Moreover, our results showed that IND and DCF slightly inhibited Beclin1 expression in both cell lines. Western blot for PRODH/POX showed the effectiveness of PRODH/POX knocking-out in MCF7 cells. In MCF7 cells IND and DCF increased PRODH/POX expression as compared to control, suggesting an increase in the proline cycle. In fact, a knockout of PRODH/POX contributed to the decrease in proline cycle enzyme PYCR1 expression, compared to control cells. However, expression of the cytosolic form of P5C reductase (PYCRL) was very low in both cell lines. Interestingly, treatment of the cells with IND and DCF inhibited COX2 expression with similar efficiency in both cell lines, suggesting that stimulation of PRODH/POX-induced apoptosis by NSAIDs could also be COX2-dependent. To establish the markers of energy state in cells, expressions of mTOR and p-AMPKα were analyzed. It was found that the studied NSAIDs decreased expression of mTOR and increased expression of p-AMPKα, which was consistent with the results on cell viability and proliferation. The increase in p-AMPKα could be a result of the shortage of α-ketoglutarate due to the lack of PRODH/POX, which converts proline to P5C, the intermediate of α-ketoglutarate. This metabolite, which enters the tricarboxylic acid (TCA) cycle, is an important energy-supporting substrate. Alternatively, it can be produced by glutamate dehydrogenase 1/2 (GLUD1/2) from glutamate. In MCF7^POX-KO^ cells, expression of GLUD1/2 was increased, compared to MCF7 cells. This suggested that in the studied conditions PRODH/POX is an important player in the energy-supporting pathway. In MCF7 cells IND and DCF increased PPARγ expression, which was considered to be a proapoptotic signal via PRODH/POX-dependent ROS generation. In MCF7^POX-KO^ cells this mechanism did not occur, as shown in Figure 3. Interestingly, PPARδ expression (which is known to support cancer growth and proliferation) was decreased in response to NSAID treatment in both cell lines. Cytosolic fraction of p53 after treatment with indomethacin and diclofenac was also decreased compared to control cells, suggesting its translocation to nuclei. Interestingly, prolidase (PEPD) expression in NSAID-treated MCF7^POK-KO^ cells was higher compared to wild-type MCF7 cells.

## 3. Discussion

The anticancer properties of NSAIDs have been known for decades [55,56,57,58]. The mechanism of this phenomenon was explained by the inhibitory effect of NSAIDs on COX2 activity during inflammation. In line with this argument is the fact that in various types of cancer (e.g., breast cancer) COX2 is overexpressed, leading to chronic inflammation supporting cell proliferation, neovascularization, and cancer growth [5,59]. Interestingly, a similar anticancer activity of NSAIDs was observed in cancer cells lacking COX2 expression and also in models of cancer cells with knockdown expression of this enzyme [8,9,60,61,62,63]. This suggested the COX2-independent anticancer mechanism of NSAIDs.

In the present study we provided evidence that NSAIDs through activation of PPARγ receptors [11] induced PRODH/POX-dependent apoptosis in MCF. This idea was based on studies describing the pro-apoptotic effect of the activated PPARγ–PRODH/POX axis [20,27,28,64,65].

PRODH/POX can, however, exert an opposite function. During conversion of proline into P5C, free electrons can support the electron transport chain producing ATP for cell survival [66,67]. However, in certain conditions free electrons can also be accepted directly by oxygen, generating reactive oxygen species (ROS). In this case ROS disturb redox balance stimulating oxidative stress, known as pro-apoptotic stimuli [28,68,69]. It seems that these opposite activities of PRODH/POX are related to the energetic state of the cell. Since cancer cells have high energy requirements and because of the Warburg’s effect glucose is not a sufficient source of energy, the cells require energy from protein degradation (mostly collagen), providing proline, among others, as a substrate for PRODH/POX resulting in ATP or ROS generation [21,22,27,28].

To evaluate the role of PRODH/POX in NSAID-induced apoptosis an innovative model of MCF7 breast cancer cells with a knockout of PRODH/POX by CRISPR/Cas9 technology was created. Unexpectedly, it was discovered that IND and DCF induced apoptosis in both cell lines (MCF7 wild-type and MCF7^POX-KO^ cells); however, the pro-apoptotic effect of NSAIDs was remarkably stronger in cells lacking PRODH/POX. The effect was accompanied by decrease in the expression of mTOR-potent cancer growth stimulator and an increase in p-AMPKα expression, a sensor of ATP level [70,71,72].

As a product of PRODH/POX activity, P5C can also be considered as an energy sensor since it can be spontaneously converted into glutamate (reversible reaction) and then to α-ketoglutarate (α-KG) by glutamate dehydrogenase (GLUD 1/2). Therefore, proline conversion into P5C can support the TCA cycle by production of its important substrate, α-KG, supporting energy production. P5C can also be produced from ornithine, a metabolite of the urea cycle [73,74,75]). We found that in MCF7^POX-KO^ cells, expression of GLUD1/2 was elevated compared to MCF7 cells. Interestingly, the studied NSAIDs stimulated the enzyme expression, suggesting that in the absence of PRODH/POX, P5C is generated from the glutamate or urea cycle, supporting TCA cycle function for energy production [27]. However, in MCF7 cells proline cycling demonstrated by increased expression of PRODH/POX in response to NSAIDs → PPARγ activation was accompanied by increased mitochondrial P5C reductase, PYCR1. In MCF7^POK-KO^ cells (in which P5C production was limited owing to PRODH/POX absence) PYCR1 expression was decreased by NSAID treatment, suggesting that it was the mechanism for limited production of pro-energetic precursors such as P5C. Simultaneously, the cancer cell growth and expression of PPARδ (energy-producing promotor) [76,77,78,79,80,81,82,83,84] was strongly inhibited by NSAIDs, enhancing the pro-apoptotic and anti-proliferative effects, particularly in MCF7^POX-KO^ cells.

Cytometric analysis showed that the population of MCF7^POX-KO^ cells contained about 12% dead cells, in comparison to about 1% dead cells in wild-type MCF7 cells, cultured in the same conditions. This showed that a lack of PRODH/POX significantly decreases cell viability. Moreover, incubation of the cells with IND and DCF resulted in a further increase in number of early apoptotic and dead cells in MCF7^POX-KO^ cells, compared to wild-type MCF7. These results were also supported by a decrease in DNA biosynthesis in NSAID-treated MCF7^POX-KO^ cells. In MCF7 cells the process was less pronounced. Similarly, collagen biosynthesis was inhibited in NSAID-treated cells as was found earlier [44,85,86,87]. However, in the case of MCF7^POX-KO^ cells, collagen biosynthesis was strongly inhibited in response to NSAID treatment. In PRODH/POX-dependent apoptosis in MCF7 cells, this activity of NSAIDs is desired because inhibition of collagen biosynthesis (the main process of proline utilization) makes proline available for PRODH/POX-dependent ROS generation and apoptosis. In MCF7^POX-KO^ cells this NSAIDs activity is also beneficial because proline carrying reducing potential is harmful for the cells [27,66,74,75,88,89]. Interestingly, in MCF7^POK-KO^ cells, the increase in intracellular proline was not only due to lack of the proline-degrading enzyme (PRODH/POX) but also because of the increase in the expression of prolidase (the proline-supporting enzyme that cleaves dipeptides containing proline). The functional significance of NSAIDs inducing the above processes was found in expression of the molecular markers of apoptosis induced in both cell lines. MCF7 cells treated with NSAIDs showed increased expression of cleaved caspase 9, which is a marker of mitochondrial apoptosis. It was accompanied by an increase in the expression of cleaved PARP, considered to be a marker of DNA damage [90,91]. DNA damage can be induced by different stress factors such as oxidative stress, which was observed in response to NSAID treatment. This was proved by fluorescence analysis of generated ROS using 2′,7′-dichlorofluorescin diacetate. These results suggested that in MCF7 cells with functional PRODH/POX, NSAIDs induced apoptosis by activating the mitochondrial (caspase 9-related) pathway in response to ROS generation as a result of PPARγ-dependent stimulation of PRODH/POX activity. In MCF7^POX-KO^ cells, the mechanism of NSAID-induced apoptosis was different. In these cells oxidative stress was not induced (ROS were not detected). This was also supported by a decrease in the expression of PPARγ, suggesting that in MCF7^POX-KO^ cells the studied NSAIDs did not induce DNA damage. Furthermore, in contrast to MCF7, in MCF7^POX-KO^ cells caspase 8 expression was significantly increased, suggesting that the extrinsic pathway of apoptosis was induced in these cells. In fact, in MCF7^POX-KO^ cells treated with IND or DCF expression of BID was more pronounced than in MCF-7 cells treated with these drugs. An increase in expression of BID can activate mitochondrial caspase 9 [92,93]. On the other hand, the studied drugs, particularly DCF, contributed to a decrease in the expression of BID in both cell lines. This effect was clearly visible in MCF-7 cells, suggesting that activation of caspase 9 in these cells could be BID-independent.

Analysis of autophagy markers (Beclin 1, Atg5, and Atg7) showed that this type of cellular death was not activated in NSAID-treated cells in both cell lines. The mechanism of NSAID-induced apoptosis in MCF7^POX-KO^ cells can be explained by the interplay of the TCA cycle, urea cycle, and proline cycle. Removing PRODH/POX from MCF7 cells influenced the concentration of P5C, the intermediate in the interconversion of proline, glutamine, α-KG, and ornithine, affecting the interplay between the metabolites of those cycles and the energy metabolism in the cells [27,94]. PRODH/POX knockout contributed to an increase in intracellular proline because of the inhibition of its degradation and the increase in prolidase expression. Moreover, in MCF7^POX-KO^ cells treated with NSAIDs collagen biosynthesis was strongly inhibited, contributing to a further increase in the concentration of intracellular proline (bearing reducing potential) affecting the redox balance in the cells. These processes are potentially involved in the mechanism of PRODH/POX-independent apoptosis in breast cancer cells. The graphical visualization of the mechanism of NSAID-induced apoptosis in MCF7^POX-KO^ cells is presented as Figure 5. Whether it is a universal mechanism in cancer remains to be established. However, based on our previous studies, the role of PRODH/POX in the apoptosis/survival of cancer cells is not only a zero–one system but rather depends on the metabolic context of the specific cell type. In our recent paper [95] we provided evidence that stimulation of PRODH/POX expression by metformin induced apoptosis in both WT and PRODH/POX knockout MCF-7 cells, but only when cultured in the absence of glutamine, while the presence of glutamine facilitated a pro-survival phenotype of the cells. Metabolomic analysis suggested that glycolysis is tightly linked to glutamine and proline metabolism in these cells, creating metabolic conditions for energy production and proline availability for PRODH/POX-dependent functions. Metformin treatment of both cell lines (WT and PRODH/POX knockout MCF-7 cells) cultured in glutamine-free medium contributed to glucose starvation, facilitating a pro-apoptotic phenotype of these cells as detected by the increase in the expression of active caspase 7 and PARP. Caspase 7 is known as an executioner protein of apoptosis activated by caspase 8 (extrinsic pathway) and caspase 9 (intrinsic pathway). The data suggested that in PRODH/POX-expressing cells and PRODH/POX knockout cells apoptosis undergoes a different mechanism. Proline availability for PRODH/POX could be important factor in the creation of a pro-apoptotic phenotype of cancer cells. In melanoma cells, characterized by intense biosynthesis of collagen (a proline-consuming process) PRODH/POX knockout did not induce apoptosis. It seems that in the melanoma cells with high capacity to utilize proline for collagen biosynthesis, the proline concentration is not enough high to induce extrinsic apoptosis. However, stimulation of PRODH/POX by metformin in melanoma cells induced ROS-dependent apoptosis, while PRODH/POX knockout abolished the effect [96]. Our other studies on MCF-7 and MDA-MB-231 cells highlighted the role of estrogens and estrogen receptors (ERα and ERβ) in PRODH/POX-dependent apoptosis [97]. We found that activation of PRODH/POX (by troglitazone) induced apoptosis only in the absence of estradiol or ERβ in MDA-MB-231 cells. Therefore, we suggest that the role of PRODH/POX in apoptosis/survival in cancer cells is metabolic context-dependent and the process occurs through various mechanisms in different types of cancer cells. Nevertheless, these results provided insight into the complexity of the molecular mechanisms creating PRODH/POX-dependent and PRODH/POX-independent apoptosis in cancer cells.

## 4. Materials and Methods

### 4.1. Materials

MCF-7 cells (ATCC^®^ HTB-22™) were obtained from ATCC (Manassas, VA, USA). MEM Eagle’s medium, fetal bovine serum, puromycin, Pen/Strep mix, and PBS were products of Gibco (Waltham, MA, USA). Propidium iodide (PI), annexin V-CF488A conjugate, and annexin V binding buffer were provided by the Biotium Company (Fremont, CA, USA). NC-Slide A2™ was purchased from Chemometec (Allerod, Denmark). Horseradish peroxidase conjugated anti-rabbit IgG and anti-mouse IgG antibodies, bacterial collagenase, 3-(4,5-dimethylthiazole-2-yl)-2,5-diphenyltetrazolium bromide (MTT), 2′,7′-dichlorofluorescin diacetate (DCFDA), indomethacin, and diclofenac were purchased from Sigma Aldrich (Saint Louis, MO, USA). Primary antibodies against COX2, p-AMPKα, mTor, caspase 8, caspase 9 total and cleaved, caspase 7 total and cleaved, BID, PARP, Beclin 1, AGT5, ATG 7, and B-actin were products of Cell Signaling Technology (Danvers, MA, USA). Primary antibodies for PRODH/POX, PPARγ, GLUD 1/2, and prolidase were products of Santa Cruz Biotechnology (Dallas, TX, USA). PYCR1, PYCRL, and PPARδ primary antibodies were obtained from Abnova (Taipei, Taiwan). Hoechst 33342 was obtained from Becton Dickinson (Franklin Lakes, NJ, USA). CRISPR All-In-One Non-Viral Vector and lipofectamine were products of Applied Biological Materials Inc. (Richmond, BC, Canada).

### 4.2. Methods

#### 4.2.1. Cell Culture

MCF7 (ATCC^®^ HTB22™) breast cancer cells were maintained in MEM Eagle’s medium, supplemented with 10% FBS, 50 U/mL of penicillin, and 50 µg/mL of streptomycin, and incubated at 37 °C in 5% CO_2_. The cells were grown on 100-mm dishes in 10 mL of complete medium. The cell culture medium was changed 2–3 times per week. For the experiments with the investigated drugs, we used MEM Eagle’s medium without FBS and Pen/Strep.

#### 4.2.2. Knockout of PRODH/POX in MCF7 Cells

A clone of the MCF7 cells with a knocked-out gene of PRODH/POX was prepared using Custom CRISPR All-In-One Non-Viral Vector, provided by Applied Biological Materials Inc. (Richmond, BC, Canada). A vector containing sgRNA sequence targeting PRODH/POX and puromycin resistance coding gene was introduced to the MCF7 cells by lipofectamine. As a selection factor puromycin in a concentration of 1.5 µg/µL was used. Antibiotic selection was performed for 10 days until about 98–100% of control cells died. After the selection period a single cell suspension was prepared, and cells were cultured on a 96-well plate to obtain single cell colonies. Efficiency of the PRODH/POX knockout was performed using the Western blot method and clone with a 100% knockout result was used for further experiments (named the MCF7^POX-KO^ cell line).

#### 4.2.3. Cell Viability Test MTT

To evaluate the cytotoxicity of the selected drugs on the MCF7 and MCF7^POX-KO^ cells, methyl thiazolyl tetrazolium (MTT) salt was used, as described in the Carmichael method [98]. This method is based on the conversion of yellow tetrazolium bromide MTT solution to the purple formazan derivatives in the live cells, and this is due to the activity of the mitochondrial dehydrogenases. For this assay, the cells were cultured in 12-well plates, at a density of 0.1 × 10^6^ cells/well. When the cells reached about 70% confluency, the culture media were removed. The wells were washed with PBS, and the fresh MEM Eagle’s media containing the studied drugs dissolved in DMSO (dimethyl sulfoxide) were added into the wells (the concentration of DMSO in the sample did not exceed 0.1%). The samples were prepared in triplicate. The cells were incubated with the compound for 24 h. The investigated drugs were in the following concentrations: 0.500 mM of indomethacin; 0.375 mM of diclofenac. After incubation, the cell culture media containing the studied drugs were removed, and the plates were washed twice with prewarmed PBS. Furthermore, the cells were incubated at 37 °C for 1 h, with MTT dissolved in PBS (0.5 mg/mL) in a volume of 1.0 mL per well. When the incubation step ended, the MTT was removed and the formazan derivatives were dissolved in the DMSO (1.0 mL per well). To quantify the formed formazan, the absorbance at 570 nm was measured using a spectrophotometer. The viability of the cells treated with the studied drugs was calculated as a percent of the control value.

#### 4.2.4. DNA Biosynthesis

To study the effect of indomethacin and diclofenac on the MCF7 and MCF7^POX-KO^ cell DNA biosynthesis, an assay of the radioactive [3H]-Thymidine incorporation into the DNA was performed. The cells were cultured on 12-well plates, as described above. When they reached about 70% confluency, they were treated with the studied drugs in the described concentrations, and 1.0 μCi of [3H]-Thymidine was added to the MEM Eagle’s medium. The samples were prepared in triplicate. The cells were incubated for 24 h at 37 °C in 5% CO_2_. The radioactivity of the samples was measured using the Tri-Carb 2810 TR Scintillation Analyzer (PerkinElmer, Waltham, MA, USA). To calculate the DNA biosynthesis, the DPMI values of the treated cells were compared to those of the control samples.

#### 4.2.5. Collagen Biosynthesis

The collagen biosynthesis was evaluated by measuring the incorporation of the radioactive 5-[3H]-Proline into the collagen. The MCF7 and MCF7^POX-KO^ cells were cultured in 12-well plates. When the cells were about 70% confluent, the culture medium was removed, and the cells were treated for 24 h with the studied drugs and 5-[3H]-Proline (5 μCi/mL). Digesting proteins, with purified *Clostridium histolyticum* collagenase, were used to determine the incorporation of the tracer into the collagen, as described by Peterkofsky et al. [99]. The radioactivity of the samples was measured by the Tri-Carb 2810 TR Scintillation Analyzer (PerkinElmer, Waltham, MA, USA). To calculate the collagen biosynthesis, the DPMI values of the treated cells were compared to those of the controls.

#### 4.2.6. SDS-PAGE and Western Blotting

For the analysis of the protein expression by Western blotting, the cells were cultured in 100 mm plates, at about 2.0 × 106 cells, and when they reached about 70–80% confluency, the culture medium was removed, and the cells were treated with the studied drugs (dissolved in culture medium). After 24 h of incubation, the culture media were removed, and the cells were harvested using a cell lysis buffer supplemented with a protease/phosphatase inhibitor cocktail. The protein concentrations of the samples were determined by the Lowry method [100]. Then, the proteins were separated using the SDS-PAGE method described by Laemmli [101]. After this step, the gels were washed in cold Towbin buffer (25 mM Tris, 192 mM glycine, 20% (*v*/*v*) methanol, 0.025–0.1% SDS, pH 8.3). The proteins in the gels were transferred onto the 0.2-µm nitrocellulose membranes by Trans-Blot (BioRad, Hercules, CA, USA). The transfer conditions were 200 mA, 3 h in freshly prepared Towbin buffer, and the temperature was maintained around 4–8 °C. The blocking of the membranes was performed by 5% NFDM for 1 h at RT. When the blocking was complete, the membranes were washed three times with 20 mL of TBS-T (20 mM Tris, 150 mM NaCl, and 0.1% Tween^®^ 20). After the washing step, the membranes were incubated with primary antibodies overnight at 4 °C. The concentration of the primary antibodies was 1:1000. Furthermore, the membranes were washed three times with 20 mL of TBS-T, and a secondary antibody conjugated with HRP solutions (1:3000) in 5% NFDM was used for 1 h at RT. Then the membranes were washed 3 times with 20 mL of TBS-T and visualized.

#### 4.2.7. ROS Formation

The cells were cultured on black wells in a 96-well plate, at 0.01 × 10^4^ cells/well. When the cells reached 70–80% confluency, the culture media were removed, the plate was washed with PBS, and 100 µL of medium containing the studied drug was added into the well. After 4 h of incubation, 0.5 µM of 2′,7′-dichlorofluorescin diacetate was added to the wells and incubated for 15 min at 37 °C in 5% CO_2_. After incubation, the culture media with DCFDA were removed and the cells were washed twice with prewarmed PBS. Then the wells were loaded with 100 µL of PBS. The cells were visualized with the BD Pathway 855 Bioimaging system in an environmental control chamber (37 °C in 5% CO_2_), and ex λ 488 nm, and em λ 521 nm. In its basic state, DCFDA is a nonfluorescent compound, and when oxidized by ROS to DCF it becomes highly fluorescent.

#### 4.2.8. Cytometric Assay for Apoptosis

The cells were cultured in 6-well plates at an initial density of 0.3 × 10^6^ cells/well. When the cells reached about 70–80% confluency, the culture media were removed. The wells were washed with PBS, and fresh MEM Eagle’s media containing the studied drugs were added into the wells. After 24 h of incubation, the cells (including floating cells) were collected by trypsinization to 1.5 mL Eppendorf tubes. The cells were centrifuged at 800× *g* for 5 min. The supernatant was removed, and the cell pellet was resuspended in 100 μL annexin V binding buffer, and then 2 μL annexin V-CF488A conjugate was added. In the next step 2 μL Hoechst 33342 (final concentration: 10 μg/mL) was added and the suspension was gently mixed by pipetting. After that, incubation of the cells at 37 °C for 15 min using a heating block was performed. Next, the stained cells were centrifuged at 400× *g* for 5 min. at room temperature, and the supernatant was removed. As the last step, the cell pellet was resuspended in 100 μL annexin V binding buffer supplemented with 10 µg/mL PI and then immediately analyzed in the NC-Slide A2. Cells with low fluorescence intensities of PI (PI negative) and low fluorescence intensities of annexin V represented living cells with high viability. The cells with high fluorescence intensities of PI (PI positive) and low fluorescence intensities of annexin V represented early apoptotic cells. The cells with high fluorescence intensities of PI (PI positive) and high fluorescence intensities of annexin V represented late apoptotic/dead cells.

#### 4.2.9. LC-MS-Based Quantitative Analysis

LC-MS analysis of proline was performed using a 1260 Infinity II high performance liquid chromatograph coupled to a 6530 Q-TOF mass spectrometer equipped with a dual electrospray ionization source (Agilent Technologies, Santa Clara, CA, USA) [102]. Chromatographic separation was conducted on a Luna HILIC column (100 × 2.0 mm, 3 μm particle size, Phenomenex, Torrance, CA, USA) maintained at 30 °C. The injection volume was 5 μL and the flow rate was 0.6 mL/min. The gradient elution of solvent A (10 mM ammonium formate in water with 0.1% formic acid) and solvent B (acetonitrile) was programmed as follows: 0–2 min with 90% B, 2–7 min with 90 to 30% B, 7–7.5 min with 30% B, 7.5–8 min with 30 to 90% B, 8–14 min with 90% B. Data were collected in positive ionization mode. The following source parameters were applied: drying gas temperature, 325 °C; drying gas flow, 12 L/min; nebulizer pressure, 45 psig; fragmentor voltage, 140 V; capillary voltage, 3000 V. The mass spectrometer operated in scan mode with a mass range of 50–1000 *m*/*z*. L-proline-d_3_ (Sigma Aldrich, St. Louis, MO, USA) was used as an internal standard. Methanol extracts of cells (containing L-proline-d_3_) were centrifuged (14,000× *g*; 4 °C; 10 min) and the supernatant was subjected to LC-MS analysis. The samples were collected in three biological repeats, injected in duplicates, and randomized before analysis. Total protein concentration was used for normalization.

#### 4.2.10. Statistical Analysis

In order to analyze the distribution of the data in individual groups, the Shapiro–Wilk test was performed. Within the normally distributed groups, the differences between the individual groups were analyzed using a one-way ANOVA, with multiple post hoc comparisons using the Bonferroni correction. In the groups where no normal distributions were shown, the nonparametric Mann–Whitney test was used. The data are presented as means ± standard error of measurement (SEM).

## 5. Conclusions

In this report we found that NSAIDs (IND and DCF) as PPARγ agonists activate PRODH/POX-induced ROS-dependent intrinsic apoptosis in breast cancer MCF-7 cells. However, in MCF7^POX-KO^ cells, NSAIDs also induced apoptosis by the ROS-independent extrinsic pathway. These results provide evidence that the studied NSAIDs induce apoptosis in both PRODH/POX-expressing and PRODH/POX knockout cells and underline the importance of PRODH/POX as a target for NSAID-induced apoptosis in breast cancer cells.

## Figures and Tables

**Figure 1 ijms-23-03813-f001:**
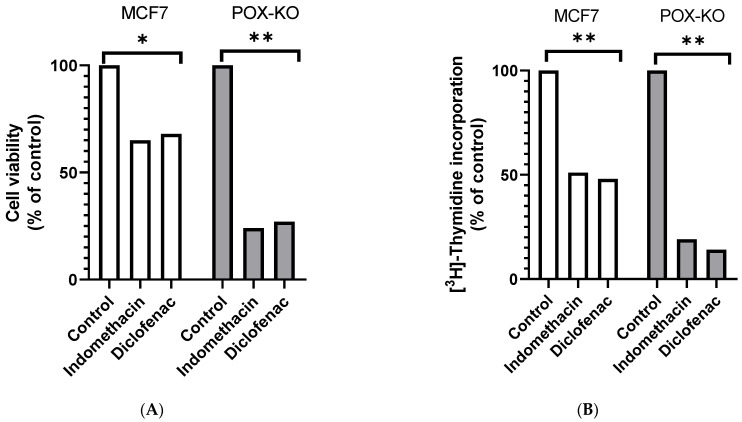
Cell viability (**A**), DNA biosynthesis (**B**), and collagen biosynthesis (**C**) in MCF-7 and MCF7^POX-KO^ cells treated for 24 h with 0.5 mM indomethacin and 0.375 mM diclofenac. Cellular proline concentration determined by HPLC-MS (**D**). The mean values ± standard error (SEM) from the 3 experiments, performed in triplicates, are presented at * *p* < 0.05 and ** *p* < 0.005.

**Figure 2 ijms-23-03813-f002:**
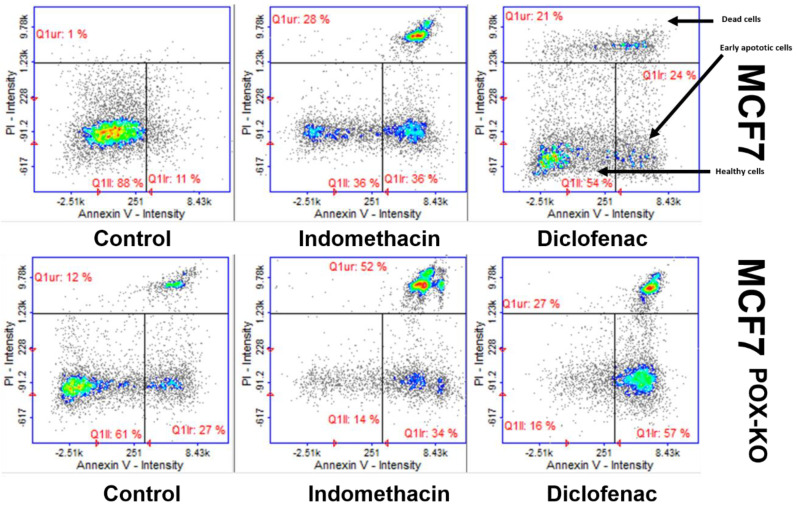
Cytometric assay of apoptosis in MCF-7 and MCF7^POX-KO^ cells. After 12 h of incubation with 0.5 mM indomethacin and 0.375 mM diclofenac, cells were stained with fluorescent dyes using annexin V and propidium Iodide.

**Figure 3 ijms-23-03813-f003:**
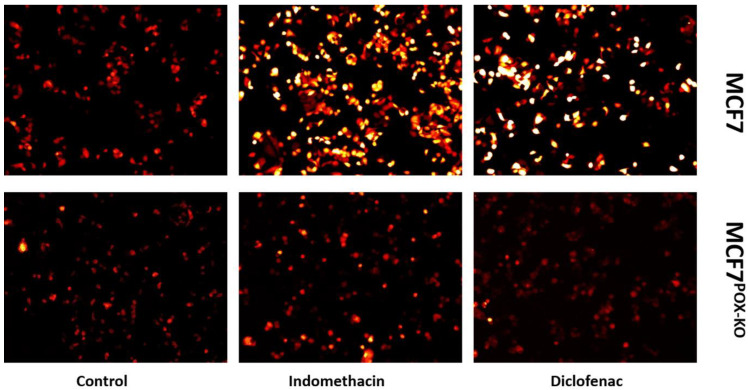
Fluorescence analysis of reactive oxygen species generation using 0.5 µM of 2′,7′-dichlorofluorescin diacetate staining. Analysis was performed on live cells using a survival chamber at −37 °C in 5% CO_2_. MCF7 and MCF7^POX-KO^ cells were incubated with 0.5 mM indomethacin and 0.375 mM diclofenac for 4 h. Red fluorescence intensity represents amount of generated ROS.

**Figure 4 ijms-23-03813-f004:**
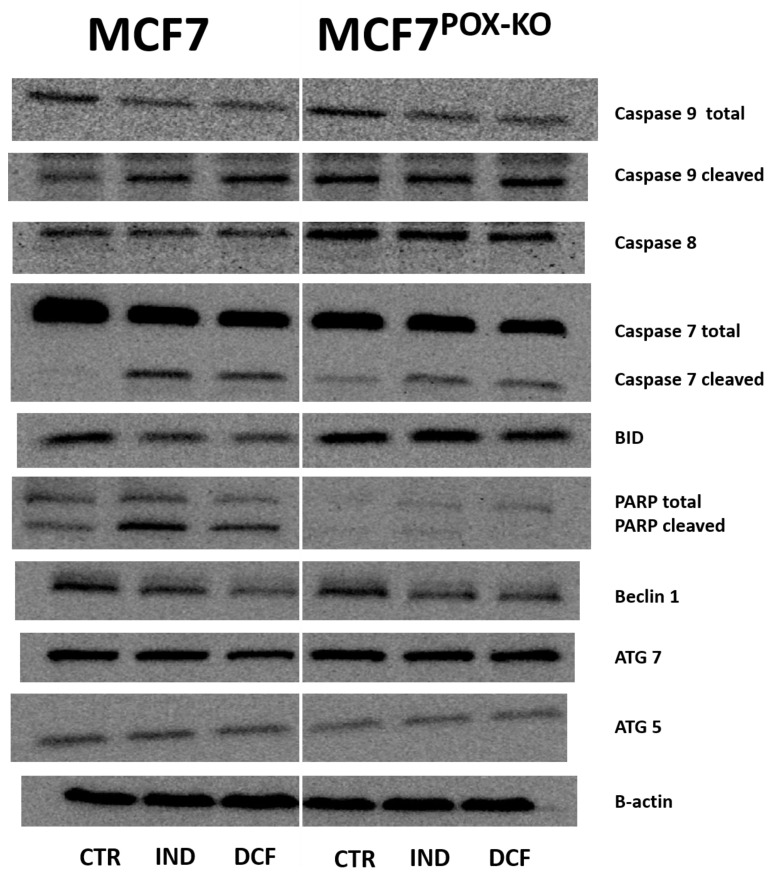
Western blot for proteins involved in PRODH/POX-dependent functions in MCF7 and MCF7^POX-KO^ cells treated for 24 h with 0.5 mM indomethacin (IND) and 0.375 mM diclofenac (DCF). For the experiment 40 µg/lane of protein lysates was used. Protein expression was normalized versus β-actin expression.

**Figure 5 ijms-23-03813-f005:**
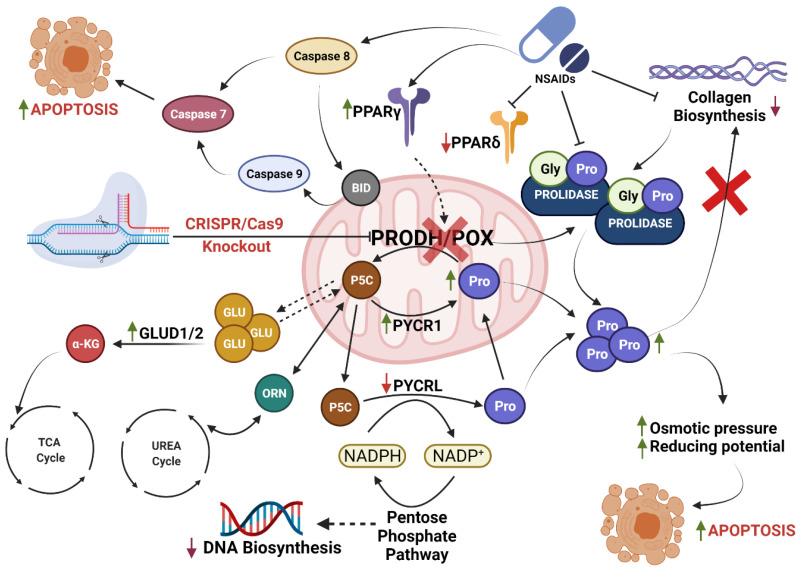
The mechanism of NSAID-induced apoptosis in MCF7^POX-KO^ cells. Created with BioRender.com, accessed on 28 January 2022, Agreement number: CP23HV3C6V.

## Data Availability

This study did not report any supporting data.

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
