# Peer review of "NSAIDs Induce Proline Dehydrogenase/Proline Oxidase-Dependent and Independent Apoptosis in MCF7 Breast Cancer Cells"

_ijms, 2022, doi:10.3390/ijms23073813_

Round 1

Reviewer 1 Report

The authors found that NSAIDs (IND and DCF) as a PPARγ agonists activate PRODH/POX–induced ROS-dependent intrinsic apoptosis in breast cancer MCF-7 cells. In addition, it also induced apoptosis by ROS-independent extrinsic pathway in MCF7POX-KO cells. 

The study provides some evidence that certain NSAIDs could induce apoptosis in both PRODH/POX expressing and PRODH/POX knock-out cells and thus PRODH/POX could serve as a target for NSAIDs-induced apoptosis in breast cancer cells.

The authors performed an overall well-conducted study, and clearly demonstrate the methods and results. It should be accepted for publicatoin.

Author Response

Response to Reviewer #1

The authors found that NSAIDs (IND and DCF) as a PPARγ agonists activate PRODH/POX–induced ROS-dependent intrinsic apoptosis in breast cancer MCF-7 cells. In addition, it also induced apoptosis by ROS-independent extrinsic pathway in MCF7POX-KO cells. 

The study provides some evidence that certain NSAIDs could induce apoptosis in both PRODH/POX expressing and PRODH/POX knock-out cells and thus PRODH/POX could serve as a target for NSAIDs-induced apoptosis in breast cancer cells.

The authors performed an overall well-conducted study, and clearly demonstrate the methods and results. It should be accepted for publication.

Response: We thank the Reviewer for evaluating of our manuscript

Reviewer 2 Report

  1. The author should consider testing their theory on more breast cancer cell lines, at least 3 cell lines of the same type.
  2. Please explain why in the figures the experiments are at different time points.
  3. Western blot figures show Cas9 as decreased expression, not increases as mentions in discussions.
  4. The same is in the case of Parp.
  5. Expression of BID is similar in western blog graphic.
  6. line 218 "studies" need to be change in "study".
  7. line 221-223 needs to rephrase.
  8. line 345 "meth-od" needs to be change in  "method".

Author Response

Response to Reviewer #2

We agree with all comments raised by the Reviewer. According to the Reviewer’s suggestion we made the following changes (red labelled sentences) in the revised manuscript:

  1. The author should consider testing their theory on more breast cancer cell lines, at least 3 cell lines of the same type.

Response: We thank the Reviewer for the comment and we understand that this issue was not adequately addressed in the present paper. In the last few years we performed studies on different cancer cell lines to establish the conditions that are required for PRODH/POX-dependent apoptosis. The information was added into “Discussion” section of revised manuscript:

We found that the role of PRODH/POX in apoptosis/survival of cancer cells is not just a zero-one system but rather depends on metabolic context of specific cell type. In our recent paper (1) we provided evidence that stimulation of PRODH/POX expression by metformin induced apoptosis in both WT and PRODH/POX knock out MCF-7 cells, however only when cultured in the absence of glutamine, while the presence of glutamine facilitated pro-survival phenotype of the cells. Metabolomic analysis suggested that glycolysis is tightly linked to glutamine and proline metabolism in these cells, creating metabolic condition for energy production and proline availability for PRODH/POX-dependent functions. Metformin treatment of both cell lines (WT and PRODH/POX knock out MCF-7 cells) cultured in glutamine free medium contributed to glucose starvation facilitating pro-apoptotic phenotype of these cells as detected by increase in the expression of active caspase-7 and PARP. Caspase-7 is known as an executioner protein of apoptosis activated by caspase 8 (extrinsic pathway) and caspase-9 (intrinsic pathway). The data suggested that in PRODH/POX expressing cells and PRODH/POX knock out cells apoptosis undergoes through different mechanism. It has been considered that proline availability for PRODH/POX could be important factor in creation of pro-apoptotic phenotype of cancer cells. In melanoma cells, characterized by intense biosynthesis of collagen (proline consuming process) PRODH/POX knock out did not induce apoptosis. It seems that in the melanoma cells with high capacity to utilize proline for collagen biosynthesis, the proline concentration is not enough high to induce extrinsic apoptosis. However, stimulation of PRODH/POX by metformin in melanoma cells induced ROS-dependent apoptosis, while PRODH/POX knock out abolished the effect (2).  Other our studies on MCF-7 and MDA-MB-231 cells highlighted the role of estrogens and estrogen receptors (ERα and ERβ) in PRODH/POX-dependent apoptosis (3). We have found that activation of PRODH/POX (by troglitazone) induced apoptosis only in the absence of estradiol or ERβ in MDA-MB-231 cells.  Therefore, we suggest that the role of PRODH/POX in apoptosis/survival in cancer cells is metabolic context dependent and the process occurs through various mechanisms in different types of cancer cells. Nevertheless, these results provided insight into complexity of molecular mechanisms creating PRODH/POX dependent and PRODH/POX independent apoptosis in cancer cells.

  1. Huynh TYL, Oscilowska I, Sáiz J, NizioÅ‚ M, Baszanowska W, Barbas C, et al. Metformin Treatment or PRODH/POX-Knock out Similarly Induces Apoptosis by Reprograming of Amino Acid Metabolism, TCA, Urea Cycle and Pentose Phosphate Pathway in MCF-7 Breast Cancer Cells. Biomolecules. 2021;11(12).
  2. Oscilowska I, Rolkowski K, Baszanowska W, Huynh TYL, Lewoniewska S, Nizioł M, et al. Proline Dehydrogenase/Proline Oxidase (PRODH/POX) Is

Involved in the Mechanism of Metformin-Induced Apoptosis in

C32 Melanoma Cell Line. IJMS. 2022;23(4)(2354):1-15.

  1. Lewoniewska S, Oscilowska I, Huynh TYL, Prokop I, Baszanowska W, Bielawska K, et al. Troglitazone-Induced PRODH/POX-Dependent Apoptosis Occurs in the Absence of Estradiol or ERβ in ER-Negative Breast Cancer Cells. J Clin Med. 2021;10(20).

  1. Please explain why in the figures the experiments are at different time points.

Response:

The analysis of various parameters required different time points. In case of the DCFDA method (the purpose of which is determination of the degree of ROS generation), an incubation time of 4 hours was used because generation of ROS is one of the first biochemical reactions inducing apoptosis in response to cytotoxic compounds. According to our experience and literature data, longer incubation could contribute to neutralization of cytotoxic compound and ROS generation in the cell. Moreover, the DCFDA probe used has a short half-life, so using it at the improper time point could disturb the results. In the case of the presented results, the incubation time with the studied compounds was determined experimentally and on the basis of literature data.

The cytometric test using Annexin V evaluates translocation and exposure of phosphatidylserine molecules on the outer side of the cell membrane. It is a process that occurs in the initial stages of apoptosis (preceding complete cell degradation). The additional use of Propidium Iodide that binds to dead cells allows the definition of 3 pools of cells: (I) healthy/alive cells,  (II) early apoptotic cells, (III) late apoptotic cells and dead cells. Performing the assay at improper time point (ie, too late) could distort the result due to the complete shift of the early and late apoptotic cell pool towards the dead cell pool. On the other hand, earlier time point could also distort the result, shifting the pools of early apoptotic or dead cells towards the pool of living cells. Since our preliminary results show that the cells treated with NSAIDs for 24 hours are dead, we experimentally set time of experiment as a 12 hours to observe apoptotic phenotype according to the purpose of the method.

In the case of MTT assay, collagen biosynthesis, DNA biosynthesis and Western Blot, 24 hours incubation of the cells with studied compounds was used on the basis of own experience, validated protocols and recommendations of the manufacturers of reagents used for these assays.

  1. Western blot figures show Cas9 as decreased expression, not increases as mentions in discussions.

Response: In Fig. 4, Cas9 was shown as a total and active (cleaved) caspase. The expression of  active Cas9 is increased in response to NSAIDs treatment only in MCF7 cells, suggesting intrinsic apoptosis.  “in MCF7 cells with functional PRODH/POX, NSAIDs induce apoptosis by activating mitochondrial (caspase 9 related) pathway in response to ROS generation as a result of PPARγ-dependent stimulation of PRODH/POX activity. In MCF7POX-KO cells, the mechanism for NSAIDs–induced apoptosis is different.”

  1. The same is in the case of Parp.

Response:  We apologize for being not precise. In the discussion we should pronounce that expression of active PARP is increased in MCF7 (wild type) cell, not in MCF7POX-KO . The following sentence was corrected and appropriate literature was added.

 “MCF7 cells treated with NSAIDs showed increased expression of caspase 9 which is marker of mitochondrial apoptosis. It was accompanied by increase in the expression of active PARP, considered as a marker of DNA damage (4, 5). DNA damage can be induced by different stress factors like oxidative stress, which was observed in response to NSAIDs treatment. It was proved by fluorescence analysis of generated ROS using 2ʹ,7ʹ-dichlorofluorescin diacetate. These results suggest that in MCF7 cells with functional PRODH/POX, NSAIDs induce apoptosis by activating mitochondrial (caspase 9 related) pathway in response to ROS generation as a result of PPARγ-dependent stimulation of PRODH/POX activity. In MCF7POX-KO cells, the mechanism for NSAIDs–induced apoptosis is different. In these cells oxidative stress is not induced (ROS are not detected). This is also supported by decrease in the expression of PPARγ suggesting that in MCF7POX-KO cells the studied NSAIDs did not induce DNA damage.

Moreover, this information is supported by paragraph in results section: ”Significant  difference in expression of total and cleaved PARP between studied cells (weak expression in MCF7POX-KO cells) suggests that in cells with functional PRODH/POX, NSAIDs-induced apoptosis is associated with DNA damage in response to environmental stress (as oxidative stress i.e.).

  1. Expression of BID is similar in western blog graphic.

Response: In the discussion section we explained  that in MCFPOX-KO cells treated with NSAIDs expression of BID was increased as compared to wild type MCF7 cells: “Furthermore, in contrast to MCF7, in MCF7POX-KO cells caspase 8 expression is significantly increased suggesting that extrinsic pathway of apoptosis is induced in these conditions. This led to increase in the expression of BID which can activate mitochondrial caspase 9”.

As it is seen on the screenshot below in MCF7POX-KO cells, Indomethacin and Diclofenac significantly increased BID expression, compared to effect found in MCF7 cell.

  1. line 218 "studies" need to be change in "study".

Response: The sentence was corrected

  1. line 221-223 needs to rephrase.

Response: The sentence was rephrased

  1. line 345 "meth-od" needs to be change in  "method".

Response: The word was corrected

We thank the Reviewer for all these insightful suggestions.

References used in response:

  1. Huynh TYL, Oscilowska I, Sáiz J, NizioÅ‚ M, Baszanowska W, Barbas C, et al. Metformin Treatment or PRODH/POX-Knock out Similarly Induces Apoptosis by Reprograming of Amino Acid Metabolism, TCA, Urea Cycle and Pentose Phosphate Pathway in MCF-7 Breast Cancer Cells. Biomolecules. 2021;11(12).
  2. Oscilowska I, Rolkowski K, Baszanowska W, Huynh TYL, Lewoniewska S, Nizioł M, et al. Proline Dehydrogenase/Proline Oxidase (PRODH/POX) Is

Involved in the Mechanism of Metformin-Induced Apoptosis in

C32 Melanoma Cell Line. IJMS. 2022;23(4)(2354):1-15.

  1. Lewoniewska S, Oscilowska I, Huynh TYL, Prokop I, Baszanowska W, Bielawska K, et al. Troglitazone-Induced PRODH/POX-Dependent Apoptosis Occurs in the Absence of Estradiol or ERβ in ER-Negative Breast Cancer Cells. J Clin Med. 2021;10(20).
  2. Abeti R, Duchen MR. Activation of PARP by oxidative stress induced by β-amyloid: implications for Alzheimer's disease. Neurochem Res. 2012;37(11):2589-96.
  3. Catalgol B, Wendt B, Grimm S, Breusing N, Ozer NK, Grune T. Chromatin repair after oxidative stress: role of PARP-mediated proteasome activation. Free Radic Biol Med. 2010;48(5):673-80.

Round 2

Reviewer 2 Report

The authors still have not correcter all the comments I have made:

  1. Recarding CASP9 line 276 -277.
  2. The expression of BID is decreased in MCF-7, and similar in MCF-7 ko
  3. Also I think the experiment should be done on more than one cell line.

Author Response

Response to the Reviewer:

The authors still have not corrected all the comments I have made:

  1. Regarding CASP9, line  2726-277.

Response: We apologize for not correcting the sentence "MCF7 cells treated with NSAIDs showed increased expression of caspase 9 which is marker of mitochondrial apoptosis. It was accompanied by increase in the expression of PARP, considered as a marker of DNA damage (91, 92).

This sentence was changed into:

“MCF7 cells treated with NSAIDs showed increased expression of cleaved caspase 9 which is marker of mitochondrial apoptosis. It was accompanied by increase in the expression of cleaved PARP, considered as a marker of DNA damage (91, 92).

  1. The expression of BID is decreased in MCF-7, and similar in MCF-7 ko.

Response: We agree that the interpretation of the expression of BID in studied cells is confusing. In the discussion section of revised manuscript we rephrased the following sentence:

“Furthermore, in contrast to MCF7, in MCF7POX-KO cells caspase 8 expression is significantly increased suggesting that extrinsic pathway of apoptosis is induced in these conditions. This led to increase in expression of BID which can activate mitochondrial caspase 9 (93, 94).”

into:

“Furthermore, in contrast to MCF7, in MCF7POX-KO cells caspase 8 expression is significantly increased suggesting that extrinsic pathway of apoptosis is induced in these cells. In fact, in MCF7POX-KO cells treated with IND or DCF expression of BID is more pronounced than in MCF-7 cells treated with these drugs. Increase in expression of BID can activate mitochondrial caspase 9 (93, 94). On the other hand, the studied drugs, particularly DCF, contributed to decrease in the expression of BID in both cell lines. This effect was clearly visible in MCF-7 cells, suggesting that activation of caspase 9 in these cells could be BID independent”

  1. Also I think the experiment should be done on more than one cell line.

Response: We agree that studies on other cancer cell lines could provide more evidence for the role of PRODH/POX in the mechanism of NSAIDs-induced apoptosis. However, we already performed such a studies on some cancer cell lines (96-98) and found that the role of PRODH/POX in apoptosis/survival in cancer cells is metabolic context dependent and the process occurs through various mechanisms in different cancer cells. It suggest that PRODH/POX-dependent apoptosis could not be an universal mechanism It depends on rate of glycolysis, presence of glutamine, estrogen receptor status, estrogens and probably several other factors. Recently we are focused on the role of collagen biosynthesis (proline utilizing process) and prolidase activity (proline supporting enzyme) in providing proline for PRODH/POX-dependent functions. Since NSAIDs were found as a potent collagen biosynthesis inhibitors it seems that in this way they may contribute to increase in concentration of proline as a substrate for PRODH/POX-dependent apoptosis in PRODH/POX expressing cells.  Whether this is the case is currently investigated. Therefore, the suggestion of the Reviewer is of great interest and will be addressed in our current and future studies.

Round 3

Reviewer 2 Report

The authors addresed all my comments.